# An Integrated Transcriptome and Proteome Analysis Reveals Putative Regulators of Adventitious Root Formation in *Taxodium* ‘Zhongshanshan’

**DOI:** 10.3390/ijms20051225

**Published:** 2019-03-11

**Authors:** Zhiquan Wang, Jianfeng Hua, Yunlong Yin, Chunsun Gu, Chaoguang Yu, Qin Shi, Jinbo Guo, Lei Xuan, Fangyuan Yu

**Affiliations:** 1Jiangsu Engineering Research Center for Taxodium Rich. Germplasm Innovation and Propagation, Institute of Botany, Jiangsu Province and Chinese Academy of Sciences, Nanjing 210014, Jiangsu, China; zhiquanjiejie@163.com (Z.W.); jfhua2009@gmail.com (J.H.); yinyl066@sina.com (Y.Y.); chunsungu@126.com (C.G.); yucg168@sina.com (C.Y.); 455244921@139.com (Q.S.); guojb0113@163.com (J.G.); 2Jiangsu Key Laboratory for the Research and Utilization of Plant Resources, Institute of Botany, Jiangsu Province and Chinese Academy of Sciences, Nanjing 210014, Jiangsu, China; 3Collaborative Innovation Center of Sustainable Forestry in Southern China, College of Forest Sciences, Nanjing Forestry University, Nanjing 210037, Jiangsu, China

**Keywords:** adventitious root formation, morphology, proteome, *Taxodium*, transcriptome

## Abstract

Adventitious root (AR) formation from cuttings is the primary manner for the commercial vegetative propagation of trees. Cuttings is also the main method for the vegetative reproduction of *Taxodium* ‘Zhongshanshan’, while knowledge of the molecular mechanisms regulating the processes is limited. Here, we used mRNA sequencing and an isobaric tag for relative and absolute quantitation-based quantitative proteomic (iTRAQ) analysis to measure changes in gene and protein expression levels during AR formation in *Taxodium* ‘Zhongshanshan’. Three comparison groups were established to represent the three developmental stages in the AR formation process. At the transcript level, 4743 genes showed an expression difference in the comparison groups as detected by RNA sequencing. At the protein level, 4005 proteins differed in their relative abundance levels, as indicated by the quantitative proteomic analysis. A comparison of the transcriptome and proteome data revealed regulatory aspects of metabolism during AR formation and development. In summary, hormonal signal transduction is different at different developmental stages during AR formation. Other factors related to carbohydrate and energy metabolism and protein degradation and some transcription factor activity levels, were also correlated with AR formation. Studying the identified genes and proteins will provide further insights into the molecular mechanisms controlling AR formation.

## 1. Introduction

Vegetative propagation is widely used for experimental and commercial propagation in forestry [1]. Among vegetative propagation methods, including budding, grafting, layering and cuttings, stem cuttings from elite genotypes is the most efficient and cost-effective method to produce large quantities of homogenous plants [2,3]. However, not all tree species are amenable to this type of propagation. With stem cutting, massive losses can occur if cuttings do not form adventitious roots (ARs), so AR formation is critical [4]. Many ecologically and economically important tree species are recalcitrant to this technique, hindering the development of large-scale plantations [5]. Studies on AR formation have focused on horticultural techniques and the process of AR formation at the physiological, anatomical and molecular levels [6,7,8]. The process of AR formation and development can be divided into three phases, induction (when molecular and biochemical changes occur), initiation (when cells begin to divide to form an internal root meristem) and expression (when the AR primordia grows and emerges from the stem) [9]. Additionally, phytohormones are critical endogenous factors in AR formation, acting directly on cell division and growth or indirectly interacting with other physiological and molecular factors [8]. The identification and characterization of the factors controlling AR formation are essential to understanding and potentially manipulating AR formation to propagate more tree species from cuttings [2].

*Taxodium* ‘Zhongshanshan’, which is called *T*. ‘Zhongshanshan’, is an interspecies hybrid clone generated from *T. mucronatum*, *T. distichum* and *T. ascendens* as three *Taxodium* species [10,11,12]. Currently, in southeastern China, *T*. ‘Zhongshanshan’ is widely used as a timber tree in river network areas, a good selection of trees for windbreak constructionin coastal region and landscape design in urban areas because it is extremely resistant to some abiotic stresses, such as highly saline environment and waterlogging [10,11,12]. Using cuttings is the main way to propagate *T*. ‘Zhongshanshan’ [13]. The rapid and healthy formation of ARs can shorten the time required for forest canopy closure and reduce the cost of management. Moreover, there are differences in capacity for AR formation among different clones during long-term breeding processes. With the increasing physiological age of clones, the capability of AR formation declines, greatly restricting the breeding and promotion of *T*. ‘Zhongshanshan’. Therefore, cultivating varieties with good growth traits, strong stress-resistance levels and a high AR-formation capacity has become the key target of *T*. ‘Zhongshanshan’ breeding. Research on optimizing cutting techniques and analyzing AR-formation characteristics has been performed [14]; however, the mechanism of AR formation remains unclear.

Biological traits are the result of the combination of genetic and environmental factors. There are various techniques to study the effects of gene expression on the regulation of AR formation [6,15]. Data of each omic have been independently analyzed to study biological processes and to explain the genetic information and metabolic pathways [16,17]. However, it is difficult to explain the regulatory biological networks of complex traits using only a single omic. Therefore, to study complex biological traits, systems biology based on multi-omics has rapidly developed [18]. Integrating multi-omics data for an analysis can compensate for problems caused by data loss, background noise and other factors found in a single omic data analysis [19]. Importantly, an analysis of combined multi-group data is more conducive to the study of models of the regulatory mechanisms of biological processes [18,19].

The central dogma of molecular biology states that DNA makes RNA and RNA makes protein. The production of intracellular proteins and the maintenance of protein concentrations require a series of closely related biological processes, including transcription, mRNA processing, degradation, translation and protein processing, modification and transport [19]. Changes in protein abundance dynamically reflect the various regulatory levels involved in gene expression. Advances in DNA sequencing and mass spectrometry have made it possible to obtain mRNA and protein abundance levels and the association analysis of proteomics and transcriptomics data is conducive to the study of the multi-level regulation of the gene expression process [20]. The expression consistency between mRNA and corresponding protein is currently believed to not be very high [21]. A joint analysis of a proteome and transcriptome will help determine the regulatory mechanism of gene expression [22].

In this study, *Taxodium* hybrid ‘Zhongshanshan 406’ (*T. mucronatum*♀ × *T. distichum*♂) (*T.* ‘Zhongshanshan 406’) was used as the experimental material for an analysis of AR-formation mechanisms at three developmental stages (initial formation of calli, primary root formation and the root-elongation period) by integrating a transcriptome analysis and proteomics. The study has important theoretical and practical significances for further breeding and for the application of new varieties of *T*. ‘Zhongshanshan’.

## 2. Results

### 2.1. Anatomical Changes During AR Formation

AR formation and development appear to consist of distinct stages, each with its own characteristics [23]. As a result, we selected four time points for analysis (Figure 1). Based on the apparent morphological characteristics of AR formation and development, cross sections of the samples (the base of cuttings (about 0.5 cm) of S0 and S1, the root tissues of S2 and S3) were examined for anatomical structural characteristics. Each stage had its own anatomical features. Both Figure 2B,C showed the features of S1, indicated that the root primordium appeared in S1, at the intersection of the phloem and cambium, at the same time white calli formed (Figure 2B,C). Figure 2C is a partial enlarged version of 2B with the root primordium. In contrast, S3 (Figure 2E) had a larger layer number of cells than S2 (Figure 2D) and aligned more closely than S2.

### 2.2. General Characterization of Transcriptome Data

To identify differentially expressed genes (DEGs) during AR formation, the expression profile was investigated using an RNA-Seq technique (accession number: PRJNA516075). A total of 105,879 unigenes with an average length of 1329 bp and an N50 of 2204 bp was obtained after de novo assembly (Appendix A). The assembled transcriptome was annotated using NT, NR, Swiss-Prot, GO, KEGG, COG and InterPro (Table 1). We determined the species distribution of NR annotation results and the top-hit species was *Picea sitchensis* (Appendix A). We also determined the functional classifications of the GO, COG and KEGG annotations as shown in Appendix A. An analysis of the annotation results of the NR, InterPro, SWISS-PROT, COG and the KEGG databases revealed that 26,150 unigenes were annotated (Appendix A).

### 2.3. DEGs During AR Formation

In this study, the PCA of all samples, shown as Appendix A, revealed a good repeatability among the biological replicates, although the partitioning of S2 and S3 was not obvious. The criteria (fold change was set as greater than 2 and the probability was set as greater than 0.8) used as the threshold to judge the significant differences in the gene expression were stringent. At all three developmental stages, using the comparison groups S1/S0, S2/S1 and S3/S2, the number of up-regulated genes was less than the number of down-regulated genes and the number of DEGs increased and then decreased following AR formation (Figure 3). To gain insights into the functional categories that were altered during AR formation, we perform GO (Appendix A) and KEGG pathway (Appendix A) classifications and functional enrichments for DEGs. In the GO analysis, a significant enrichment (*p* ≤ 0.05) of DEGs was found in ‘biological processes’ (BPs), ‘cellular components’ (CCs) and ‘molecular functions’ (MFs). During the formation phase of calli and root primordia, the enrichment of the extracellular region in CCs, the carboxy-lyase activity in MFs and the oxidation-reduction process in BPs were the highest. The most enriched categories during primary root formation were the hydrogen peroxide catabolic process in BPs, the extracellular region in CCs and oxidoreductases activity in MFs. The DEGs in the root growth stage were mainly concentrated in peroxidase activity in MFs and cellular homeostasis in BPs. To further determine which biological pathways were significantly regulated during AR development (*p* ≤ 0.05), a KEGG pathway enrichment analysis was performed for the three developmental stages. In the S1/S0 stage, DEGs were enriched in glycolysis/gluconeogenesis and other metabolic pathways. In the S2/S1 stage, the biosynthesis of secondary metabolites contained a large number of DEGs. The S3/S2 stage was significantly enriched in metabolic pathways.

### 2.4. qRT‑PCR Confirmation of Selected Gene Expression Levels

To validate the transcript profiles produced in this study, 12 genes were randomly selected for qRT-PCR analysis (Figure 4). The expression patterns detected by qRT-PCR for these 12 genes were consistent with those in the profiles and the average of the correlation was 0.79, which indicated our RNA-Seq data were reliable (Figure 4 and Appendix A).

### 2.5. Proteomic Analysis of DEPs During AR Formation

To identify the differentially expressed proteins (DEPs) during AR formation, the samples used for RNA-Seq (i.e., S0, S1, S2 and S3) were also used for the proteomics analysis (accession number: PXD012834). A total of 23,032 unique peptides were identified (Appendix A). These 23,032 peptides were matched to 7356 unique protein groups in 8 samples. The variation in expression during AR formation is shown in Figure 5. A total of 1076 specifically expressed proteins, 536 being up-regulated and 540 down-regulated, were significantly identified in a comparison of S1/S0. For the comparison of S2/S1, 996 proteins were identified as up-regulated, while 889 proteins were down-regulated. In the comparison of S3/S2, 609 proteins were down-regulated, while 435 were up-regulated. The *CV* value showed a good repeatability (Appendix A) [24]. Annotation analyses by GO, COG and KEGG were implemented based on the identified proteins (Appendix A). The GO enrichment analysis showed the GO terms enriched for DEPs represented important or typical biology functions (Appendix A). The results showed that the chloroplast of CCs, the oxidoreductase activity of MFs and photosynthesis of BPs were the most enriched categories in the stage of calli and root primordia formation. The most abundant categories in primary root formation stage were related to photosynthesis in BPs, chloroplast in CCs and hydrolase activity in MFs. The DEPs in the root growth stage were mainly concentrated on the nucleosome of CCs, oxidoreductase activity, which acts on the sulfur groups of donors, of MFs and nucleosome assembly of BPs. Because proteins usually interact with each other to play roles in certain biological functions, we performed a pathway-enrichment analysis of the DEPs based on the KEGG database during AR formation (Appendix A). The DEPs were largely enriched in metabolic pathways during the S1/S0 and S2/S1 stages and the phenylpropanoid biosynthesis pathway was significantly enriched during the subsequent S3/S2 stage.

### 2.6. Conjoint Analysis of DEPs and DEGs During AR Formation

The conjoint analysis of DEGs and DEPs during AR formation and development was performed. Transcripts were detected for 99.5% of the proteins (Figure 6). Based on the quantitative analysis of their expression changes, DEPs and their corresponding genes were used for the conjoint analysis. The relationship between the numbers of proteins and genes is shown in Figure 6. To investigate the concordance between differential expression levels of transcript and protein, we created scatterplot of the expression ratio of each comparison group. The scatter plot analysis shows the log2 fold change of the corresponding protein: mRNA (Figure 7). We divided the associate protein: mRNA into 9 types using 9 quadrants, respectively representing the differential expression trend of protein: mRNA. As shown in Figure 7, almost all of the corresponding protein: mRNAs were concentrated at the center of the plot, where protein and mRNA levels did not vary above 1.2- and 2- fold, respectively. In addition to this position, eight quadrants were found in which either the mRNA or protein levels exceeded the level of variation (Figure 7). We observed 624, 1117 and 88 concordant dots, representing a positive correlation between protein abundance and transcript accumulation in different stage (S1/S0, S2/S1 and S3/S2) of AR formation (3 and 7 in Figure 7). This study focused on the analysis of these proteins/genes, including GO and KEGG pathway enrichment analyses.

The most highly enriched GO terms during AR formation were metabolic, cellular and single-organism processes in BPs; membrane, cell, cell part and organelle in CCs; and catalytic activity and binding items in MFs. Consistent with the dots in quadrants 3 and 7, a KEGG pathway enrichment was revealed at both the transcript and protein levels during the three developmental stages. At the S1/S0 stage, 21 KEGG orthologs (KOs) were significantly down- or up-regulated, including indole alkaloid biosynthesis, phenylpropanoid biosynthesis, flavonoid biosynthesis, cutin, suberine and wax biosynthesis, photosynthesis-antenna proteins, photosynthesis, carbon fixation in photosynthetic organisms and carbon metabolism (Appendix A). Additionally, at the S2/S1 stage, 29 KOs were significantly down- or up-regulated, including phenylpropanoid biosynthesis, DNA replication, other glycan degradation, glutathione metabolism, phenylpropanoid biosynthesis, betalain biosynthesis and vitamin B6 metabolism (Appendix A). When analyzing the S3/S2 stage, five KOs, photosynthesis-antenna proteins, photosynthesis, cyanoamino acid metabolism, 2-oxocarboxylic acid metabolism and glucosinolate biosynthesis, were significantly down- or up-regulated (Appendix A). Thus, significant metabolic changes occurred during the AR-formation period. There were several proteins/genes involved in the key pathways of indole alkaloid biosynthesis, peroxisome and the tricarboxylic acid (TCA) cycle were up-regulated during S1/S0 and S2/S1 and down-regulated during S3/S2 (Figure 8). Some proteins/genes involved in ubiquitin-mediated proteolysis were down-regulated during the AR-formation period (Figure 8). In addition, 21 shared up/down regulated proteins/genes were related to plant hormone signal transduction, which is a key pathway in AR formation (Figure 9). In the signal transduction pathways of hormones, such as auxin, cytokinin and gibberellins, most proteins/genes were up-regulated in the S1/S0 and S2/S1 stages and the proteins/genes at upstream of the transduction pathway were mainly highly expressed in the S1/S0 stage, while the downstream proteins/genes were mainly expressed in the S2/S1 stage.

### 2.7. Identification of TFs During AR Formation

To identify TFs involved in AR formation, we carried out a targeted analysis of TFs from all DEGs and DEPs. We found 14 TFs (3 GRAS, 3 NAC and 8 WRKY) that were regulated in S1/S0, S2/S1 and S3/S2 (Figure 10). In total, 9 and 4 were up-regulated in S1/S0 and S2/S1, respectively. And 3 TFs were down-regulated in S3/S2 (Figure 10).

## 3. Discussion

We selected four time points based on morphological and anatomical changes [9]. In this study, we performed a conjoint transcriptome and proteome analysis to identify DEGs and DEPs during AR formation and development. We identified many DEGs in the comparisons of S1/S0, S2/S1 and S3/S2, respectively. The PCA indicated the good reproducibility of the transcriptome data, although the partitioning of S2 and S3 was not obvious. It may be that molecular regulation was not as high in S3/S2 as in other stages and the sampling parts of S2 and S3 were the same. Based on the reference transcriptome, several proteins were identified as regulated. And transcripts were detected for 99.5% of the proteins. Based on the quantitative expression changes determined in the study, DEPs and their corresponding genes were used in a conjoint analysis. Compared with S2/S1 in Figure 7, more corresponding protein: mRNAs in S0/S1 and S3/S2 were concentrated at the center of the plot, where protein and mRNA levels did not vary above 1.2- and 2- fold. The reason for this quantitative difference may be that more genes regulating AR formation tend to be differentially expressed in S2/S1. As shown in Figure 7, several mRNA: protein ratios during AR formation were found to fall in quadrants 1, 2 and 4, in which the mRNA: protein ratios reflected high levels and these genes may be regulated at the post-transcriptional or translational level and inhibit protein translation. Additionally, substantial mRNA: protein ratios were found in quadrants 6, 8 and 9, in which the mRNA: protein ratios reflected low levels and these genes may express accumulated proteins that are regulated at the post-transcriptional or translational level. Only a few mRNA: protein ratios reflected significant positive changes at both the transcript and protein levels. The genes here were unregulated or less regulated at the post-transcriptional or translational level and the measured results of the regulation of these genes were more credible [19]. To decipher the molecular processes related to AR formation, it is meaningful to analyze both of the transcriptomic and proteomic data. The integrative transcriptomic and proteomic data helped us identify a set of proteins/genes that could be involved in AR formation.

### 3.1. Plant Hormones in AR Formation

There is already growing evidence that the development of AR is related with the regulation of endogenous factor and/or signaling [25]. Among the endogenous factors, plant hormones are critical in AR formation through their interactions with other physiological and molecular factors [8]. In the analysis of AR formation, auxin and ethylene are considered to be activators, on the contrary, gibberellins is often described as inhibitors but some opposite effects have been observed in some studies [26]. Some of the regulated genes and proteins related to plant hormones are shown in Figure 9. Among these, the gibberellin pathways may play positive role in AR formation and the regulation may function in combination with that of other hormones, including auxin and ethylene. These data depict a detailed picture of the regulatory network involved in hormone signal transduction.

Auxin is a class of phytohormones that are widely used in plant propagation to induce root formation in cuttings [27,28]. The levels of the proteins encoded by the auxin-responsive gene *GH3* in *Arabidopsis* are positively correlated with the number of ARs [29]. Additionally, auxin stimulated AR formation by inducing the expression of the *GH3* family [30]. In this study, *GH3* (CL11043.Contig1_All) was significantly up-regulated at the protein level during the initiation formation of the callus and finally breakthrough outside the organization to form root. Thus, this gene may be positively involved in the formation of AR in *T*. ‘Zhongshanshan’. We also observed that upstream proteins/genes in biological processes, such as indole alkaloid biosynthesis, significantly changed from S0 to S2. Auxin is involved in every aspect of root development in both monocotyledons and dicotyledons, from cell fate acquisition to meristem initiation, emergence and elongation [25]. Our results showed that auxin plays a key role in the S0 to S2 stage in *T*. ‘Zhongshanshan’. Thus, exogenous auxin can be used to stimulate AR formation and development in *T*. ‘Zhongshanshan’.

Other hormones interact with auxin signaling to regulate AR development [6]. As a class of plant growth regulators, cytokinin has a positive effect for cell division and development of shoot. Cytokinin is a kind of inhibitor for auxin and then suppressesthe formation of AR in many species, such as *Arabidopsis* and *Populus* [31,32]. Despite this, low concentrations of cytokinin play active roles in the initial stages of AR induction in apple and Monterey pine cuttings [33,34]. In this study, *CRE1* (CL2837.Contig10_All), a membrane-located receptor of cytokinin signals, was identified and up-regulated in the early stages of AR initiation. Furthermore, *AHP* (Unigene10480_All), a mediator in a multistep phosphorelay pathway for cytokinin signaling, was also up-regulated [35]. Thus, cytokinins may play important and positive roles in the induction of the root primordia and calli in *T*. ‘Zhongshanshan’.

Ethylene promotes AR formation by increasing auxin levels [6]. Ethylene insensitive 3 (*EIN3*) acts as a positive regulator downstream of the ethylene signal transduction pathway. *EIN3* encodes a TF in *Arabidopsis* and it works downstream of *EIN2* and upstream of *AtERF1*, an early ethylene responsive gene [36]. In our study, *EIN3* was up-regulated during primary root formation at both the mRNA and protein levels, indicating that ethylene signal transduction may promote primary root formation, which would be used to improve horticultural rooting techniques. Additionally, *CTR1* (CL395.Contig13_All) was up-regulated in the S1/S0 stage and significantly down-regulated in the S2/S1 stage, suggesting that the gene plays a positive role in the initial calli formation [37].

Gibberellins also appear to interact with auxin during AR development and act synergistically with ethylene to elevate the number of penetrating roots and the growth rate of emerged roots, as well as the AR length [6,38]. DELLA proteins belong to plant-specific GRAS family of TFs, are participatedin the recognition ofgibberellins signaling and the interactionof gibberellins and the gibberellins receptor GIBBERELLIN INSENSITIVE DWARF 1 (GID1) [39]. The binding of gibberellins and GID1 inducesthe formation of the GAGID1-DELLA complex and plays a keyrole inplant growth and root elongation [39]. *PIF3* (CL2016.Contig1_All) is regulated by gibberellins signaling and plays an important role in plant growth and development [40]. The PIF family is highly conserved in terms of domain structure but each *PIF* gene likely has its own unique biological role [41]. Here, at the mRNA level, the gene encoding GID1, which was upstream of the gibberellins signal-transduction pathway, was up-regulated during the S1/S0 and S2/S1 stages. Additionally, the genes of the DELLA family and *PIF3*, which were located downstream, were only up-regulated at the S1/S0 stage. However, at the protein level, the genes encoding GID1 also showed upward trends in the S1/S0 and S2/S1 stages and the DELLA family of genes and *PIF3* were up-regulated in the S2/S1 stage. The DELLA family of genes and *PIF3* may be regulated at the post-transcriptional or translational level and their encoded proteins may accumulate, which influences the morphological development of ARs.

Brassinosteroids are natural plant-growth promoting products widely distributed in the plant kingdom [42]. Both auxin and brassinosteroids induce a lot of auxin-signaling genes which related to root growth and development [43]. Brassinosteroids significantly increase the rooting capacity of Norway spruce cuttings but whether interact with auxin during AR formation is not clear [44]. Combination of BRI1 receptor and brassinosteroids can stimulate the interaction of BRI1 and a related BAK1 receptor and then forming of a complex of BRI1/BAK1 [45,46]. Then the glycogen synthase kinase 3/SHAGGY (GSK3/SHAGGY) serine/threonine kinase encoded by *BIN2*, which is a negative regulator of brassinosteroids signaling, will be inhibited and the brassinosteroids-responsive genes, as *BZR1*, will be activated [45,46]. We found that genes encoding the BRI1 receptor were down-regulated at the S3/S2 stage at both the mRNA and protein levels. However, the regulatory models of the genes in S1/S0 and S2/S1 were different in this study. This may be the result of differences in the gene functions and the specific functions of these genes needs further study.

### 3.2. Identification of Other Potential Regulators Involved in AR Formation

Protein complexes corresponding to polyphenol oxidase, per-oxidase and indole acetic acid oxidase emerge in the early stages of rooting of *Phaseolus aureus* [47]. However, in contrast, Upadhyaya (1986) claimed that polyphenol oxidase and per-oxidase were not involved in root initiation but rather in their development [48]. In fact, the involvement of oxidative enzymes in AR formation has been abundantly described in the literature, with frequently contradictory results and thus appeared to be species- or cultivar-dependent. In this study, we found a gene (Unigene32249_All, annotated with peroxiredoxin) related to the antioxidant system in the peroxisome pathway that was significantly up-regulated at both the mRNA and protein levels during the initiation and formation of calli and the breakthrough to form roots. Thus, oxidative enzymes may be involved in root initiation. Consequently, we could not exclude the possibility that the peroxisome pathway might be important during AR formation. The exact roles of the oxidative enzymes during AR formation needs to be investigated further.

Ubiquitin-mediated protein modifications play important roles in many cellular signal-transduction pathways [49]. After activation by ubiquitin-activating enzyme, ubiquitin was passed on to the ubiquitin-conjugating enzyme and created links with a target substrate with the help of the ubiquitin-protein ligase [50]. In *Arabidopsis*, auxin/indole-3-acetic acid (Aux/IAA) protein degradation was triggered by a ubiquitin-protein ligase. Then, an increase in the degradation of Aux/IAA proteins led to a higher concentration of active auxin factors, which activated the transcription of auxin-responsive elements, resulting in higher transcription levels of auxin-responsive genes [51,52]. During AR formation and development in this study, the expression levels of specific genes involved in the ubiquitin-mediated proteolysis pathway (Unigene10039_All, CL9098.Contig3_All and CL2764.Contig3_All) were down-regulated. The genes may have initiated the process of AR formation by degrading certain inhibitory factors related to auxin homeostasis and its intricate signaling network.

The nutrients consumed during AR formation and development were main carbohydrates and nitrogen compounds, providing nutrition and energy sources [53]. The TCA cycle is the most important physiological process, being the ultimate metabolic pathway of the three major nutrients (carbohydrates, lipids and amino acids) and the hub of carbohydrate, lipid and amino acid metabolism. Because of poor correlations, only 5 DEGs related to the TCA cycle, which were positively correlated to DEPs, were identified during AR formation. In the process of the conversion from citrate to isocitrate, aconitate hydratase plays a key role in the TCA cycle. The gene encoding aconitate hydratase (CL2325.Contig4_All) was up-regulated during the S1/S0 stage. The gene may play an important role in resource redistribution during AR formation. Additionally, several other proteins related to carbohydrate and energy metabolism, protein degradation and photosynthesis were also correlated with AR formation. The results indicated that increasing the levels of stored nutrients and metabolic activities may promote root primordium initiation [54,55]. Additionally, the down-regulation of genes or proteins that are linked to the TCA cycle, such as the gene encoding 2-oxoglutarate dehydrogenase E1 component (Unigene21704_All), maybe responsible for the light hypersensitivity related to photosynthesis during AR formation [54].

### 3.3. TFs in AR Formation

Among transcripts, significant expression change levels were detected in putative genes encoding TFs in 35 TF families at the successive stages of AR formation in poplar [56]. In this study, the TFs with significant expression changes in both mRNA and protein levels during AR formation and development mostly belonged to the GRAS, NAC and WRKY families.

GRAS family, including SCARECROW (SCR), SCARECROW-LIKE (SCL) and SHORT-ROOT (SHR), was involved in the earliest stages of AR formation in the presence of exogenous auxin [57,58]. And that some studies found the expression of *SHR* in root, root primordiaand even rooting-competent cells without the application of exogenous auxin [58]. Expressions of TFs related to GRAS (Unigene884_All, Unigene7692_All and CL5775.Contig5_All) were highly induced during initiation formation of callus. The study highlighted an important role for GRAS in the formation of root primordia, consistent with previous research [59].

The NAC family contains plant-specific TFs. *NAC1* is induced by auxin to promote lateral root development and is regulated by the ubiquitin degradation system in *Arabidopsis* [60]. In addition, *MIR164* interacts with *NAC1* to down-regulate auxin signals related to lateral root formation [61], providing a mechanism for homeostatic balance and preventing the over-proliferation of roots [62]. NAC may also play a role during AR development in *T*. ‘Zhongshanshan’. In this study, *NAC* genes were up-regulated during the initiation of AR primordia. These *NAC* genes maybe downstream responsive genes in the auxin-signaling pathway. They were regulated by *MIR164* at the transcriptional level and by the ubiquitin degradation system at the protein level.

As a large gene TF families, the members of WRKY play important roles in regulating various signaling pathways of plant development, including the germination, dormancy and development of seeds, plant development and responses to biotic and abiotic stresses [63]. Many WRKY factors, which in plants play main roles in the innate immune systems, are concerned with transcriptional reprogramming associated with plant defense responses [64]. From leaving the mother seedling to becoming a mature plant, the relevant regulatory factor-mediated mechanical repair of wounds and defense responses against external damage were necessary for cuttings. 8 WRKY TF-encoding genes were found to be differentially expressed during AR development. Several genes were up-regulated at the initiation of calli formation in cuttings at the protein level. These genes may be involved in the wound repair and defense responses in cuttings.

## 4. Materials and Methods

### 4.1. Plant Materials

Softwood cuttings of *T.* ‘Zhongshanshan 406’ were collected from the Institute of Botany, Jiangsu Province & Chinese Academy of Sciences, Nanjing, China. One-year-old healthy softwood cuttings were selected for the experiment from mother seedlings that were less than 8 years of age. Each cutting was cut into a length of 15–18 cm and one-half of the leaves on each cutting were removed to prevent excessive moisture loss. To prevent mildew and other fungal attacks, 1000 mg/L carbendazim was used to thoroughly spray the seedbed (containing moistened perlite: peat soil with organic matter, 1:1) in a ventilated greenhouse under normal growth conditions (approximately 30°C) and a photoperiod of 14/10 h of light/dark. After seedbed treatment, the cuttings were planted individually in the seedbed. The conventional management of cuttings was then adopted. Based on apparent morphological changes and anatomical structural characteristics, samples at three pivotal time points were used as experimental samples. The examination of the anatomical structural characteristics was carried out using a scanning electron microscope (Quanta 200, FEI Company, Hillsboro, Oregon, USA) at an acceleration of 10–15 kV. The first time point (S1) was the initial formation of calli, the second (S2) was status when the primary root formed and the third (S3) was the root-elongation period (Figure 1). The control time point (S0) was taken at 0 day, which was the dormant cortex period and the cuttings were stored immediately after excision (Figure 1). The tissues for others were frozen immediately in liquid nitrogen and stored at −80 °C until needed.

### 4.2. RNA Isolation, Illumina Sequencing and Raw Data Processing

Three independent biological replicates, each consisting of 20 randomly selected stem cuttings, were taken at each time point. The total RNA of each sample was extracted using an RNAprep Pure Plant Kit (Polysaccharides- & Polyphenolics-rich; Tiangen, Beijing, China) according to the manufacturer’s instructions [65]. The concentration and quality of the RNA were determined using a NanoDrop 2000 spectrophotometer (Thermo Scientific, Waltham, MA, USA) and 2% gel electrophoresis was used to determine the quality and integrity of the total RNA. The construction of the cDNA library and sequencing were performed as previously reported with minor modifications [66,67]. Briefly, mRNA was enriched from total RNA using oligo (dT) magnetic beads. Following purification, the mRNA was fragmented into small pieces under elevated temperature. Then, the first-strand cDNA was synthesized using the mRNA fragments as templates. The double-stranded cDNA was synthesized with buffer, dNTPs, RNase H and DNA polymerase I, purified with a QiaQuick PCR Purification Kit (Qiagen, Hilden, Germany) and washed with EB buffer for end repair and single nucleotide A (adenine) addition. Finally, sequencing adaptors were ligated to the fragments. The acquired fragments were purified by agarose gel electrophoresis and enriched by PCR amplification. The library products were ready for sequencing using IlluminaHiSeq™ (Illumina, San Diego, CA, USA). The resulting data were analyzed by a slightly modified version of the procedure as previouslydescribed [68,69,70,71]. For all libraries, raw sequencing reads of low quality, adapter/primer contaminants and duplicated reads were removed, after which clean reads of high quality were obtained. Reads from all 12 samples were concatenated and a reference assembly was created using the Trinity software package (version 2.0.6, https://trinityrnaseq.github.io/). The assembled sequences were called unigenes and the Tgicl software package (version 2.0.6, http://sourceforge.net/projects/tgicl/files/tgicl) was used to remove spliced and redundant sequences to acquire non-redundant unigenes that were as long as possible.

### 4.3. Functional Annotation of the Transcriptome

The unigenes were aligned with sequences in the NCBI non-redundant nucleotide sequences (NT), NCBI non-redundant protein sequences (NR), Swiss-Prot (a manually annotated and reviewed protein sequence database), Kyoto Encyclopedia of Genes and Genomes (KEGG) and Cluster of Orthologous Groups of proteins (COG) using a BLAST algorithm-based search (version 2.2.23, http://sourceforge.net/projects/tgicl/files/tgicl) [72]. Blast2GO (version 2.5.0, https://www.blast2go.com) and NR annotation results were used for the gene ontology (GO) annotation [73]. Then, InterProScan5 (version 5.11-51.0, https://code.google.com/p/interproscan/wiki/Introduction) was used for the InterPro annotation [74]. Based on the priority order of the functional annotations of NR, SwissProt, KEGG and COG, we selected the best aligned fragments as the coding sequences of the unigenes. ESTScan software 55 (version 3.0.2, http://sourceforge.net/projects/estscan) was used to determine the coding regions and sequence orientation when a unigene could not be aligned to any of the databases [75]. Getorf (version EMOBOSS: 6.5.7.0, http://genome.csdb.cn/cgi-bin/emboss/help/getorf) and hmmsearch (version 3.0, http://hmmer.org) were used to detect open reading frames and map them to the domains of transcription factor (TF) proteins (data from PlntfDB) [76,77].

### 4.4. Differential Expression Analysis

Clean reads were mapped to unigenes with Bowtie2 (version 2.2.5, http://bowtie-bio.sourceforge.net/Bowtie2/index.shtml). Then, the gene expression levels were estimated by RSEM (version 1.2.12, http://deweylab.biostat.wisc.edu/RSEM) for each sample [78]. A principal component analysis (PCA) was performed with all samples using princomp, a function of EdgeR. We set S1/S0, S2/S1 and S3/S2 as the comparison groups. The differentially expressed genes (DEGs) were selected by the method of NOIseq, with the fold change set at greater than 2 and the probability set at greater than 0.8 [79]. Additionally, classification and functional enrichments of DEGs by GO and KEGG pathways were performed using phyper, another function of Edge R (*p* ≤ 0.05).

### 4.5. Quantitative Real-Time PCR (qRT-PCR) Validation of DEGs

qRT-PCR was performed to validate gene expression. cDNA was synthesized from 1.0 μg of RNA using a PrimeScript RT Kit with gDNA Eraser (TaKaRa, Dalian, China) according to the manufacturer’s protocol. qRT-PCR was conducted in 96-well plates and performed on the Analitik Jena qTOWER2.2 PCR System (Biometra, Gottingen, Germany) using the following cycling conditions: 50°C for 2 min, 95°C for 10 min and 40 cycles of 95°C for 15 s and 60°C for 1 min, followed by a melting curve analysis in which the PCR products were heated from 60 to 95°C. Each reaction mix contained 2 µL previously diluted cDNA (1:3), 10 µL FastStart Universal SYBR Green Master (ROX; Roche Applied Science, Mannheim, Germany), 6.8 µL RNase-free water and 6 pmol each primer, for a final volume of 20 µL. Primers for the reference gene adenine phosphoribosyltransferase (*APRT*) (accession no. KX431853) and random selected genes were designed using the Oligo 6.0 software [11]. Each sample was analyzed in triplicate and all of the primers are listed in Appendix A.

### 4.6. Protein Extraction and Isobaric Tags for Relative and Absolute Quantitation (iTRAQ) Reagent Labeling

The plant materials used for the iTRAQ analysis were the same as those for RNA sequencing (RNA-Seq). Two independent biological replicates were taken at each time point. Protein was extracted from each sample according to the method of Yang [80]. The protein concentration and quality were determined using the Bradford method and confirmed by 12% sodium dodecyl sulfate-polyacrylamide gel electrophoresis (SDS-PAGE) [81]. The iTRAQ analysis was carried out as previously reported with slight modifications [80]. Briefly, the quantified proteins were digested 12 h with trypsin at 37°C. The tryptic peptides were labeled with the 8-plex iTRAQ reagents (AB Sciex, Foster City, CA, USA) following the manufacturer’s protocol. The samples were independently labeled with iTRAQ tags 113, 114, 115, 116, 117, 118, 119 and 121. A Shimadzu LC-20AB liquid-phase system was used to separate peptides in a Gemini C18 column (4.6 × 250 mm). High-efficiency separation was achieved using a Shimadzu LC-20AD high-performance liquid chromatography (HPLC) directly attached to a mass spectrometer.

### 4.7. Liquid Chromatography Linked to Tandem Mass Spectrometry (LC-MS/MS) Analysis, Protein Identification and Quantification

Data acquisition was performed using a TripleTOF 5600 System (SCIEX, Framingham, MA, USA) fitted with a Nanospray III source (SCIEX, Framingham, MA, USA) and a pulled quartz tip as the emitter (New Objectives, Woburn, MA, USA). The high-sensitivity mode was set for survey scans. Raw data files were transformed into MGF files using ProteoWizard tool and the exported MGF files were searched by Mascot 2.3.02 (Matrix Science, Boston, MA, USA). NCBInr and Swiss Prot/UniProt database searches were performed for protein identification. In addition, the transcriptome database was useful for protein identification. The automated IQuant software quantitatively analyzed the labeled peptides with isobaric tags [82]. We used the ratio of the standard deviation sigma to the mean (*CV*) to evaluate the reproducibility. The lower the *CV* value, the better the reproducibility. In this project, we also set S1/S0, S2/S1 and S3/S2 as comparison groups. *p* values, representing the probability that the protein is differentially expressed, of less than 0.05 and fold change ≥ 1.2 were set as the significant thresholds for differential expression. Then, we searched against the GO, COG and KEGG databases to classify and identify differentially expressed proteins (DEPs). Additionally, classification and functional enrichments for DEPs by GO and KEGG pathways were performed (*p* ≤ 0.05).

### 4.8. Conjoint Analysis of Transcriptomic and Proteomic Data

To investigate the concordance between transcript and protein levels, we created scatterplots of the expression ratios (log 2 fold change) of each comparison group. Then, we searched against the GO and KEGG databases to classify and identify DEGs and DEPs having different expression patterns. Significant pathway enrichment was examined using the hypergeometric test and the significance was set at *p* ≤ 0.05.

## 5. Conclusions

In summary, an integrated analysis based on the transcriptome and proteome was performed during AR formation and development. The DEGs and DEPs were identified. Hormonal signal transduction was different at different developmental stages and we speculated that using the corresponding hormone combination exogenously at a proper concentration may stimulate AR formation and development on stem cuttings. Additionally, several other factors related to carbohydrate and energy metabolism, protein degradation and some TFs also correlated with AR formation. The genes and proteins we have identified will provide valuable insight into the molecular mechanisms controlling AR formation.

## Figures and Tables

**Figure 1 ijms-20-01225-f001:**
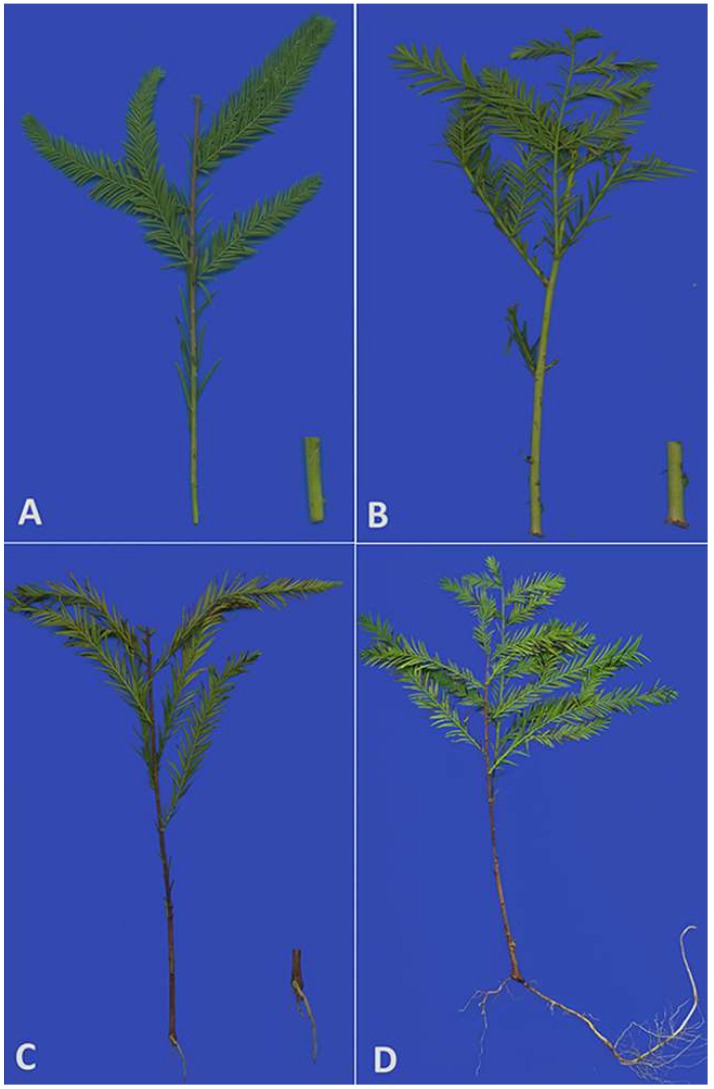
Adventitious root formation in *T.* ‘Zhongshanshan 406’ at different developmental time points: (**A**) dormant cortex period (S0); (**B**) the initial formation of calli (S1); (**C**) the formation of the primary root (S2); and (**D**) the root-elongation period (S3).

**Figure 2 ijms-20-01225-f002:**
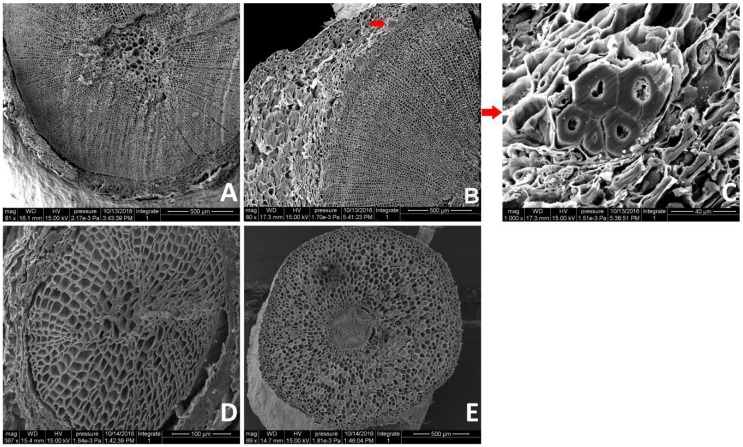
Anatomical changes during AR formation: (**A**) dormant cortex period (S0); (**B**) the initial formation of calli (S1); (**C**) a partial enlarged version of 2B with the root primordium at the intersection of the phloem and cambium (S1); (**D**) the formation of the primary root (S2); and (**E**) the root-elongation period (S3). The two red arrows point to the location of the root primordium.

**Figure 3 ijms-20-01225-f003:**
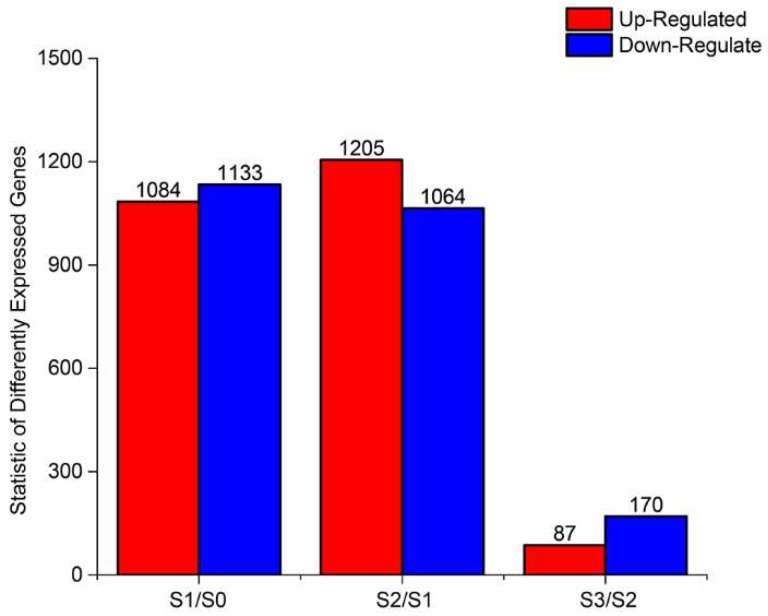
The statistics of DEGs.

**Figure 4 ijms-20-01225-f004:**
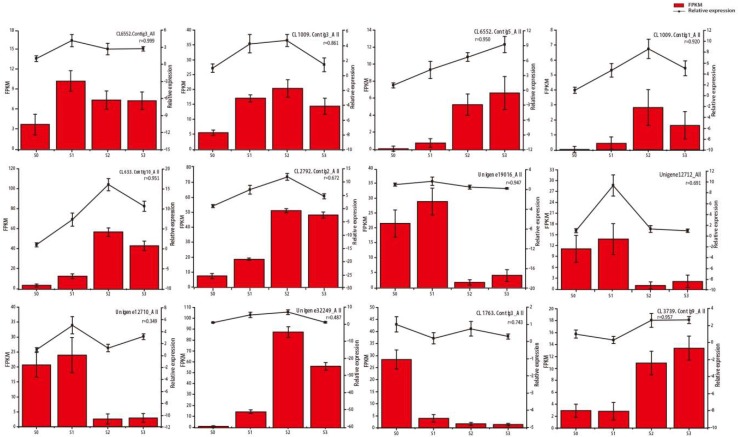
Quantitative real-time PCR (qRT-PCR) validation of differential gene expression. Histograms were fragments per kilobase of transcript per million fragments mapped (FPKM) detected by RNA-Seq, line graphs were relative expression validated by qRT-PCR, r represents the correlation coefficient between two methods.

**Figure 5 ijms-20-01225-f005:**
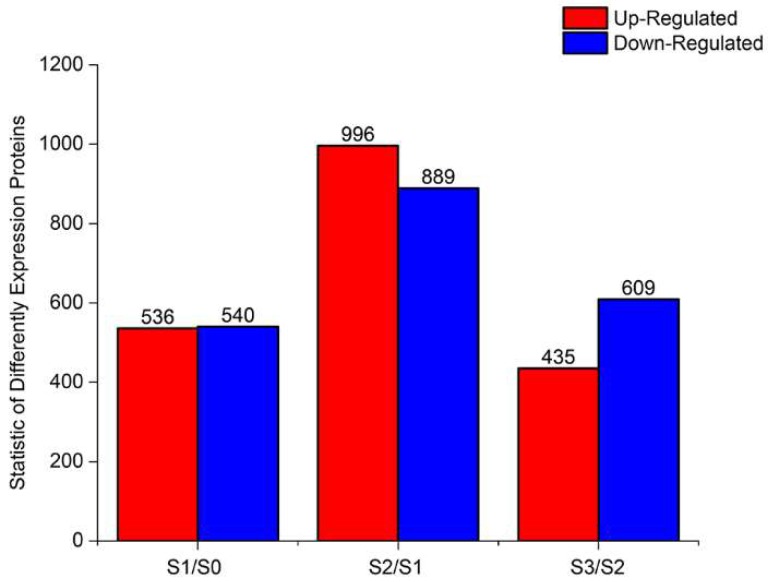
The statistics of DEPs

**Figure 6 ijms-20-01225-f006:**
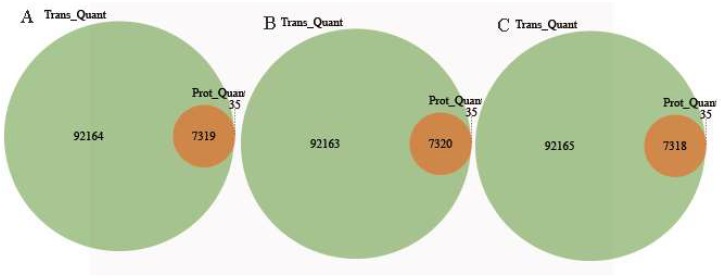
Congruency of the detected transcriptome and proteomein each different stage (A, S1/S0; B, S2/S1 and C, S3/S2).

**Figure 7 ijms-20-01225-f007:**
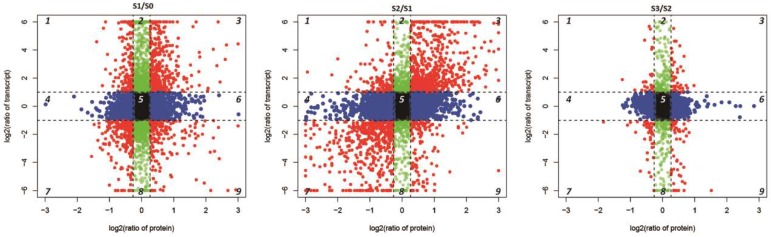
Comparison of changes in protein and cognate mRNA abundance levels. The relative changes in abundance (S1/S0, S2/S1 and S3/S2) are shown using a log2 scale. The associate protein: mRNA were divided into 9 types using 9 quadrants and several colors, respectively representing the differential expression trend of protein: mRNA. The center of the black plot was marked as quadrant 5, where protein and mRNA levels did not vary above 1.2- and 2- fold, respectively. The red dots in quadrants 3 and 7, representing a positive correlation between protein abundance and transcript accumulation in different stage. The red dots in quadrants 1 and 9 show negative correlations. Green dots in quadrants 2 and 8 indicate no difference in protein expression, while transcriptional expression tends to be up-regulated or down-regulated. Blue dots in quadrants 4 and 6 show differences in protein expression, while transcriptional expression tends to be no significant difference.

**Figure 8 ijms-20-01225-f008:**
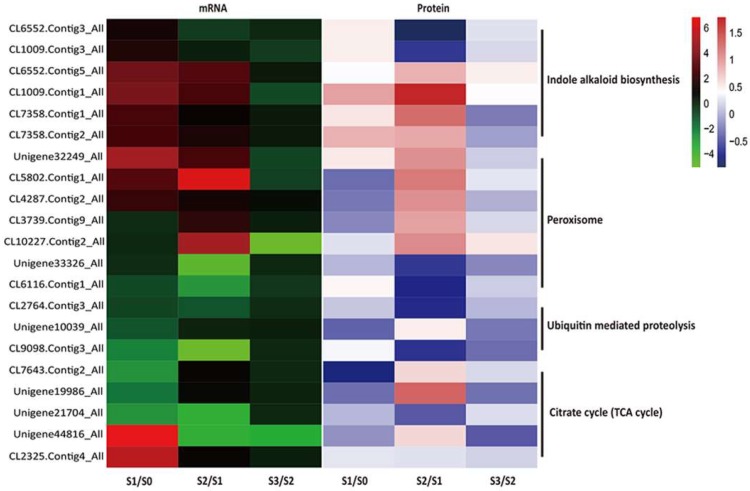
Analysis of differentially abundant proteins and genes in relevant pathways. Red color indicates an increase; green and blue color indicate decreases. The relative changes in abundance (S1/S0, S2/S1 and S3/S2) are shown using a log2 scale.

**Figure 9 ijms-20-01225-f009:**
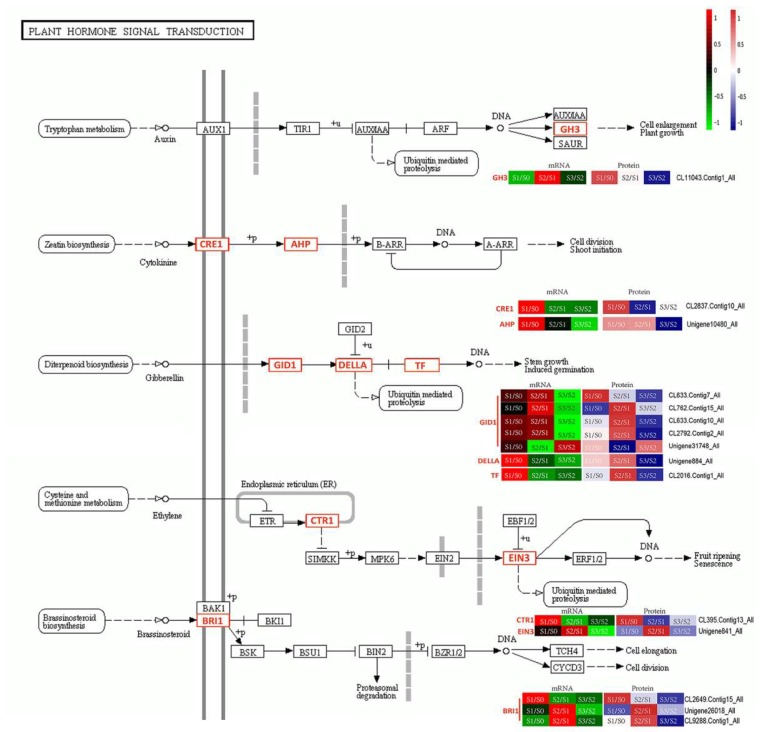
The key pathways during adventitious root formation. The relative changes in abundance (S1/S0, S2/S1 and S3/S2) are shown using a log2 scale. Double lines, cytomembrane; Grey wide dotted line, nuclear membrane; Grey wide line, endoplasmic reticulum membrane; +u, ubiquitination; +p, phosphorylation; 
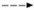
, indirect effect; 
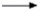
, activation; 
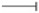
, inhibition; 
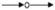
, expression; 
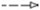
, indirect effect; 
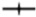
, dissociation.

**Figure 10 ijms-20-01225-f010:**
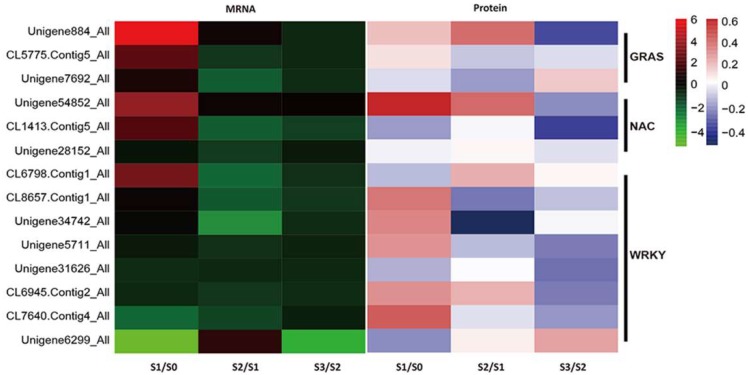
Analysis of differentially abundant TFs during AR formation and development. The relative changes in abundance (S1/S0, S2/S1 and S3/S2) are shown using a log2 scale.

**Table 1 ijms-20-01225-t001:** Summary of the functional annotation results.

Values	Total	Nr Annotated	Nt Annotated	Swissprot Annotated	KEGG Annotated	COG Annotated	Interpro Annotated	GO Annotated	Overall
Number	105,879	68,046	48,467	51,068	53,682	32,668	55,196	23,381	71,637
Percentage	100%	64.27%	45.78%	48.23%	50.70%	30.85%	52.13%	22.08%	67.66%

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
