# Peer review of "An Integrated Transcriptome and Proteome Analysis Reveals Putative Regulators of Adventitious Root Formation in Taxodium ‘Zhongshanshan’"

_ijms, 2019, doi:10.3390/ijms20051225_

Round 1
Reviewer 1 Report
I the paper by Wang and colleagues the authors generated and meta-analyzed the transcriptome and proteome data of Taxodium ‘Zhongshanshan’, and interspecies hybrid widely used in Chinese agriculture as a timber, windbreak and landscape tree. The study has important theoretical and practical significance for further breeding.
The bioinformatical study is well designed and performed. However the results in general are not as outstanding as they could be, because the experimental design is not perfect.
The authors chose the AR developmental stages that are too late to make any conclusions. Stages S0 and S1 are fine (despite S1 is a heterogenous tissue yet), but stages S2 and S3 are two late and the difference between S2 and S3 is not so big (as clearly seen from generated genome-wide data).
If the authors managed more detailed anatomical analysis and collected the samples more carefully for particular stages of adventitious root primordium development, than the work would be of much more interest.
My suggestion is to remove the data and analysis on S3 phase, than the comparison of molecular events on S0-2 looks better. Than all figures should be adjusted in accordance. Especially Figure 2, which has to be supplemented with subscriptions (especially on B and C).
The data on RNA-Seq results validation with the help of qRT-PCR does not look convincing for half of the genes. It is better to estimate correlation between RNA-seq and qPCR data and present it as scatterplots for comparison of S0/S1 and S1/S2 fold changes (separately).
The data on functional annotation was present for p-value <0.05, but it should be done with q-value.
Conjoint analysis of DEP and DEG does not look convincing as well. Yes, there are many causes why mRNA level and the protein level does not correlate. But when an experimental validation is absent for proteins there is a high chance that the disagreement is not only in molecular machinery, but also because of the problems in the experiment.
Thus, before comparison between the developmental stages, the authors must compare the transcriptome and the proteome at each stage and show that there is a significant overlap in the expressed mRNAs and proteins. As an example, see Figure 1 here: https://www.mcponline.org/content/mcprot/early/2012/07/25/mcp.M112.020461.full.pdf
Maybe an adjustment would be in need before the conjoint analysis.
On the Figure 7 we see no tendencies in correlation of DEPs and DEGs on the left and the right plot, but the central plot suggest that there is a tendency because areas 3, 5, 7 are more abundant.
It is better present Fig 3 in supplementary
Fig. 4 – grammar mistakes
There is no link to the raw experimental data
Author Response
Response to Reviewer 1 Comments
I the paper by Wang and colleagues the authors generated and meta-analyzed the transcriptome and proteome data of Taxodium ‘Zhongshanshan’, and interspecies hybrid widely used in Chinese agriculture as a timber, windbreak and landscape tree. The study has important theoretical and practical significance for further breeding.
The bioinformatical study is well designed and performed. However the results in general are not as outstanding as they could be, because the experimental design is not perfect.
The authors chose the AR developmental stages that are too late to make any conclusions. Stages S0 and S1 are fine (despite S1 is a heterogenous tissue yet), but stages S2 and S3 are two late and the difference between S2 and S3 is not so big (as clearly seen from generated genome-wide data).
If the authors managed more detailed anatomical analysis and collected the samples more carefully for particular stages of adventitious root primordium development, than the work would be of much more interest.
Point 1: My suggestion is to remove the data and analysis on S3 phase, than the comparison of molecular events on S0-2 looks better. Than all figures should be adjusted in accordance. Especially Figure 2, which has to be supplemented with subscriptions (especially on B and C).
Response 1: Thank you for your comments! And I am sorry I did not explain clearly the issue about the chosen AR developmental stages. In this study, based on the previous results and apparent morphological changes, we selected four time points. The first time point (S1) was the initial formation of calli, the second (S2) was status when the primary root formed, and the third (S3) was the root-elongation period. The control time point (S0) was taken at 0 d, when the cortex was dormant. Then we divided AR formation and development into three stages, including callus and root primordium formation stage (S0-S1), primary root formation stage (S1-S2) and root elongation stage (S2-S3). The methods of sample collection followed these three papers (Brinker et al., 2004; Wang et al., 2016; Steffens et al., 2006).
Among of them, the base of cuttings (about 0.5 cm) of S0 and S1, the root tissues of S2 and S3 were taken for anatomical structural characteristics (Figure 2). In contrast, S3 (Figure 2E) had a larger layer number of cells than S2 (Figure 2D) and aligned more closely than S2.
257 genes and 1044 proteins were differentially expressed in the root elongation periods (S3/S2). By conjoint analysis, it was found that 88 concordant dots putting on a positive correlation between protein abundance and transcript accumulation at this stage. For example, genes involved in the key pathways of indole alkaloid biosynthesis, peroxisome, ubiquitin-mediated proteolysis pathway, the tricarboxylic acid (TCA) cycle and some TF families (GRAS, NAC and WRKY) were down-regulated at the S3/S2 stage at both the mRNA and protein levels. The elongation growth of root system plays an important role in the survival rate and seedling quality of cuttings, so these identified genes and proteins at this stage also have theoretical and practical significance for further breeding. For all these reasons, we selected four time points in the present study.
We have added more explanations and relative information in the revised manuscript (lines 96-103).
If experts and editors insist that S3 is redundant, please review to us again, we will remove relevant trials.
References
Brinker M, van Zyl L, Liu W, et al. Microarray analyses of gene expression during adventitious root development in Pinus contorta[J]. Plant physiology, 2004, 135(3): 1526-1539.
Wang P, Ma L, Li Y, et al. Transcriptome profiling of indole-3-butyric acid-induced adventitious root formation in softwood cuttings of the Catalpa bungei variety ‘YU-1’at different developmental stages[J]. Genes & Genomics, 2016, 38(2): 145-162.
Steffens B, Wang J, Sauter M. Interactions between ethylene, gibberellin and abscisic acid regulate emergence and growth rate of adventitious roots in deepwater rice[J]. Planta, 2006, 223(3): 604-612.
Point 2: The data on RNA-Seq results validation with the help of qRT-PCR does not look convincing for half of the genes. It is better to estimate correlation between RNA-seq and qPCR data and present it as scatterplots for comparison of S0/S1 and S1/S2 fold changes (separately).
Response 2: Thank you for your comments! We have added the correlations between RNA-seq and qPCR data in Figure 4 in the revised manuscript. We also showed the average of the correlation in the revised manuscript (lines 151-153). And the catterplots for comparison were showed in Figure S9 (supplementary material).
Point 3: The data on functional annotation was present for p-value <0.05, but it should be done with q-value.
Response 3: Answer: Thank you for your comments! And I am sorry I did not explain clearly the issue. We have checked the research report and recent papers (Lei et al., 2018; Quan et al., 2017). According to the annotation results and official classification, we used the phyper in R software for enrichment analysis and classified the functions of differentially expressed genes for p-value <0.05. p value is calculated as showed in the word file in the attachment.
So, the data on functional annotation was present for p-value < 0.05.
References:
Wikihttps://en.wikipedia.org/wiki/Hypergeometric_distribution
Lei C, Fan S, Li K, et al. iTRAQ-based proteomic analysis reveals potential regulation networks of IBA-induced adventitious root formation in apple[J]. International journal of molecular sciences, 2018, 19(3): 667.
Quan J, Meng S, Guo E, et al. De novo sequencing and comparative transcriptome analysis of adventitious root development induced by exogenous indole-3-butyric acid in cuttings of tetraploid black locust[J]. BMC genomics, 2017, 18(1): 179.
I am so sorry that I had made a writing mistake in method of protein quantification, and I have corrected the mistake in the modified version (line 514).
Point 4: Conjoint analysis of DEP and DEG does not look convincing as well. Yes, there are many causes why mRNA level and the protein level does not correlate. But when an experimental validation is absent for proteins there is a high chance that the disagreement is not only in molecular machinery, but also because of the problems in the experiment.
Response 4: Thank you for your comments! Future research is necessary to validate the proteins, but technical imperfection is a difficult problem for perfecting this experiment now. We will do our best to perfect this study in the future. According to your suggestion, we have compared the transcriptome and the proteome at each stage and showed the significant overlaps in the expressed mRNAs and proteins in Figure 6 in the revised manuscript (line 200).
Point 5: Thus, before comparison between the developmental stages, the authors must compare the transcriptome and the proteome at each stage and show that there is a significant overlap in the expressed mRNAs and proteins. As an example, see Figure 1 here: https://www.mcponline.org/content/mcprot/early/2012/07/25/mcp.M112.020461.full.pdf
Response 5: Thank you for your comments! Before comparison among of the developmental stages, we have added the information of comparation between the transcriptome and the proteome at each stage. The significant overlaps in the expressed mRNAs and proteins were showed in Figure 6 in the revised manuscript. And the result showed that transcripts were detected for 99.5% of the proteins. Since Table 2 and Figure 6 showed the same result in the revised version, we deleted Table 2 and retained Figure 6, which expressed the data more directly.
If experts and editors feel that Table 2 is meaningful, please review to us again, we will add it again.
Point 6: Maybe an adjustment? would be in need before the conjoint analysis.
Response 6: I am sorry I did not explain clearly the issue. Based on the reference transcriptome, several proteins were identified as regulated. And transcripts were detected for 99.5% of the proteins. Then DEPs and their corresponding genes were used in a conjoint analysis. The relationship between the numbers of proteins and genes is shown in Figure 6 in the revised manuscript. To investigate the concordance between differential expression levels of transcript and protein, we created scatterplot of the expression ratios of each comparison group. The scatter plot analysis showed the log2 fold changes of the corresponding protein: mRNA for comparisons of S0/S1, S1/S2 and S3/S2 (Figure 7). We divided the associate protein: mRNA into 9 types using 9 quadrants, respectively representing the differential expression trend of protein: mRNA. Only a few mRNA: protein ratios reflected significant positive changes at both the transcript and protein levels. The genes here were the research emphases.
We have added some relative details in the revised manuscript (lines 185-192).
Point 7: On the Figure 7 we see no tendencies in correlation of DEPs and DEGs on the left and the right plot, but the central plot suggests that there is a tendency because areas 3, 5, 7 are more abundant.
Response 7: I am sorry I did not explain clearly the issue. The scatter plot analysis showed the log2 fold changes of the corresponding protein: mRNA for comparisons of S0/S1, S1/S2 and S3/S2 (Figure 7). We divided the associate protein: mRNA into 9 types using 9 quadrants, respectively representing the differential expression trend of protein: mRNA. Compared with S2/S1, more corresponding protein: mRNAs in S0/S1 and S3/S2 were concentrated at the center of the plot, where protein and mRNA levels did not vary above 1.2- and 2- fold. The reason for this quantitative difference may be that more genes regulating AR formation tend to be differentially expressed in S2/S1. Although S1/S0 in the early stage and S3/S2 in the later stage were also the processes of the formation and development of AR, the numbers of DEGs and DEPs were less than S2/S1, which is reflected in the Figure 7 with less abundance in areas 3, 7 and other areas. However, there are still some genes and proteins with positive correlation and negative correlation. We have added some relative details about Figure 7 in the revised manuscript (lines 253-259), so that readers can better understand the meaning of the result.
Point 8: It is better present Fig 3 in supplementary
Response 8: Thank you for your comments! We have moved Figure 3 in supplementary as Figure S6.
Point 9: Fig. 4 – grammar mistakes
Response 9: I am so sorry that I had made a mistake here, and I have corrected the mistake in the modified version (line 148).
Point 10: There is no link to the raw experimental data
Response 10: Thank you for your comments! We have added the link to the raw experimental data in the modified version (lines 115 and 160).
Reviewer 2 Report
The work of Wang and colleagues studies the gene expression and protein content profile of a commercially important plant, Taxodium‘Zhongshanshan’ during vegetative propagation.
The studies on tree plants present challenges as for instance the long generation time and the difficult genetic manipulation. The use of the two powerful tools, trascriptomics and proteomics, allowed a relative detail view of the pathways involved in the adventitious root formation.
The clear description of the results contributes to an easy reading for a wider audience.
The discussion and conclusions have a right degree of speculation about the main biological pathway that contributes to the progress of the stages of adventitious root formation after cutting.
I consider that this work can provide an important reference point not only for commercial use but also in the field of plant organogenesis.
Author Response
Response to Reviewer 2 Comments
Point 1: The work of Wang and colleagues studies the gene expression and protein content profile of a commercially important plant, Taxodium‘Zhongshanshan’ during vegetative propagation.
The studies on tree plants present challenges as for instance the long generation time and the difficult genetic manipulation. The use of the two powerful tools, trascriptomics and proteomics, allowed a relative detail view of the pathways involved in the adventitious root formation.
The clear description of the results contributes to an easy reading for a wider audience.
The discussion and conclusions have a right degree of speculation about the main biological pathway that contributes to the progress of the stages of adventitious root formation after cutting.
I consider that this work can provide an important reference point not only for commercial use but also in the field of plant organogenesis.
Response 1: Thank you for your comments!
Round 2
Reviewer 1 Report
I found the authors explanation and the manuscript improvements are enough for understanding. I still have some concerns to the experimental design and to the comparison of DEGs and DEPs, but I agree that this would need much more work. The manuscript has novelty and some results are important, thus i can suggest it for publishing.
Minor comments:
The y axis on figures 3,5 contain grammar mistakes, and the legend for these images should be extended
Please make the subscriptions on the figure 6 more carefully.
Author Response
Response to Reviewer 1 Comments
Comments and Suggestions for Authors
I found the authors explanation and the manuscript improvements are enough for understanding. I still have some concerns to the experimental design and to the comparison of DEGs and DEPs, but I agree that this would need much more work. The manuscript has novelty and some results are important, thus i can suggest it for publishing.
Minor comments:
Point 1: The y axis on figures 3,5 contain grammar mistakes, and the legend for these images should be extended
Response 1: I am so sorry that I had made a mistake here, I have corrected the mistakes in the modified version (lines 147 and 183). And thank you for your comments! The legends for Figure 3 and 5 have been extended in the modified version (lines 148 and 184).
Point 2:Please make the subscriptions on the figure 6 more carefully.
Response 2: Thank you for your comments! We have added more explanations and relative information on Figure 6 in the revised manuscript (lines 189-193).